# Composite fermion liquid to Wigner solid transition in the lowest Landau level of zinc oxide

D. Maryenko[1], A. McCollam[2], J. Falson[3,4], Y. Kozuka[3,5], J. Bruin[2,4], U. Zeitler[2] & M. Kawasaki[1,3]

Interactions between the constituents of a condensed matter system can drive it through a plethora of different phases due to many-body effects. A prominent platform for it is a dilute two-dimensional electron system in a magnetic field, which evolves intricately through various gaseous, liquid and solid phases governed by Coulomb interaction. Here we report on the experimental observation of a phase transition between the composite fermion liquid and adjacent magnetic field induced phase with a character of Wigner solid. The experiments are performed in the lowest Landau level of a MgZnO/ZnO two-dimensional electron system with attributes of both a liquid and a solid. An in-plane magnetic field component applied on top of the perpendicular magnetic field extends the Wigner-like phase further into the composite fermion liquid phase region. Our observations indicate the direct competition between a composite fermion liquid and a Wigner solid formed either by electrons or composite fermions.

[1] RIKEN Center for Emergent Matter Science (CEMS), Wako 351-0198, Japan. [2] High Field Magnet Laboratory (HFML-EMFL) and Institute for Molecules and Materials, Radboud University, 6525 ED Nijmegen, The Netherlands. [3] Department of Applied Physics and Quantum-Phase Electronics Center (QPEC), The University of Tokyo, Tokyo 113-8656, Japan. [4] Present address: Max Planck Institute for Solid State Research, Stuttgart, Germany. [5] Present address: National Institute for Materials Science, Tsukuba, Ibaraki, Japan. Correspondence and requests for materials should be addressed to D.M. (email: maryenko@riken.jp)

A magnetic field $B$ applied perpendicularly to a two-dimensional charge carrier system modifies its density of states and places the charge carriers on a ladder of discrete Landau levels (LL). The Coulomb interaction between the charged particles acting on the magnetic length scale $l_B = \sqrt{\hbar/eB}$ can be tuned by varying the magnetic field strength. Thereby, the high mobility carriers evolve through the various correlation phases[1]. When the electrons occupy half of available states in the lowest LL, e.g., filling factor $\nu = 1/2$, the electrons prefer to reduce their interaction by virtue of capturing two magnetic flux quanta resulting in the emergence of new particles, called composite fermions (CF)[2,3]. These particles form a Fermi surface at $\nu = 1/2$ and move in an effective field $B_{\text{eff}} = B - B_{\nu=1/2}$ (Fig. 1, middle panel) giving rise to magnetoresistance oscillations. At even lower filling factors a Wigner solid, a crystalline phase of charged particles (electrons or CF) driven by the repulsive Coulomb force and yet another manifestation of many-body correlations, emerges as a ground state of the electron system (Fig. 1).

Being in the lowest Landau level (LL) the electron system experiences competition between the composite fermion liquid phase and the magnetic field induced Wigner solid phase, which manifests as a large magnetoresistance peak around or below $\nu = 1/3$[4,6]. A liquid-solid transition may follow the Kosterlitz–Thouless model, whereas the particles can form a hexatic phase characterized by bond-oriented nearest-neighbor ordering[7–11]. An intermediate phase of the liquid-solid transition may also take the form of microemulsion phases associated with a liquid crystalline phase[11–13]. Departing from the liquid phase of CF at $\nu = 1/2$, a

formation of both a composite fermion Wigner solid and phase transition to intermediate phases may appear feasible. The idea of realizing a composite fermion Wigner solid was put forward in a number of theoretical works[14–19]. Recent experiments focusing on GaAs-based 2DES have been gradually accumulating evidence pointing towards the realization of CF Wigner solid[20–24]. Intuitively the CF crystal is stabilized when the CF of nearby liquid states release two of their vortices to stabilize the crystal, whereas the undressed particles retain their energetically favorable correlations[18]. Thus a two-flux CF crystal borders the four-flux composite fermion liquid phase, whereas an electron crystal phase is embedded in two-flux composite fermion liquid and forms between filling factors $\nu = 1/3$ and $\nu = 2/5$ for a high enough LL mixing[19]. Thus the transition between the liquid and the solid can be highly non-trivial and is realized in the lowest LL of a two-dimensional charge carrier system by the transformation of the underlying particle type.

Here, we study the magnetotransport in a ZnO heterostructure (see: Methods) in the magnetic field region between the CF liquid phase formed at $\nu = 1/2$ and the high resistivity phase appearing at higher field and exhibiting attributes of a Wigner solid[25,26]. LL mixing, the ratio between electron–electron interaction energy and the cyclotron energy, is 4.2 at $\nu = 1/2$ in this heterostructure and the magnetotransport in the region of interest features a character of both CF liquid and crystalline phase. The presence of such a region with the interlaced character highlights a non-trivial nature of phase transition, the details of which can further be masked by the inhomogeneous potential landscape arising from inevitable crystallographic disorder. The transition between the two phases can be tuned by the application of an in-plane magnetic field. As a result of the phase intermixture, the state at filling factor $\nu = 1/2$ can be formed by a composite fermion liquid and some intermediate state arising in the course of liquid-solid transition. Owning to a simple band structure of ZnO as compared to p-type GaAs, this oxide material is an attractive platform to access the competition between liquid and solid phases in the fractional quantum Hall regime.

## Results

**Experiment in perpendicular magnetic field.** Figure 2 shows a full scan of the magnetotransport from 0 T to 33 T applied perpendicular to the 2DES plane. Several fractional quantum Hall states are observed around $\nu = 3/2$, consistent with previous results[26] and, in addition, developing minima are observed at $\nu = 9/5$, $12/7$, $9/7$, and $6/5$. Furthermore, up to six fractional quantum Hall states are observed on both sides of $\nu = 1/2$. Close inspection of the transport around $\nu = 1/2$ reveals a distinct asymmetry; $R_{xx}$ maxima between fractional quantum Hall states for $\nu < 1/2$ are much larger than those for $\nu > 1/2$. This increase becomes increasingly dramatic between $\nu = 2/5$, $1/3$, and $2/7$. Such a high resistance phase between $\nu = 1/3$ and $2/7$ has also been observed in GaAs, and was interpreted as the electron Wigner solid pinned by disorder. Two mechanisms have been identified for the appearance of the Wigner solid around these filling factors: one is a large LL mixing, which modifies the ground state energies of fractional quantum Hall states and Wigner solid;[4,5,19] the other is short-range disorder[4,27,28]. Both mechanisms are more pronounced in ZnO-heterostructures than in GaAs[25,29].

In Fig. 2 the apparent high resistance phases are colored and marked as HRP1, HRP2, and HRP3. On the basis of Wigner solid studies in other materials system, the characteristics of HRP1, HRP2, and HRP3 are typical of the Wigner solid. The temperature dependence of $R_{xx}$ for HRP1, HRP2, and HRP3 resembles the melting of the Wigner solid (Supplementary Fig. 1a). The non-linear current-voltage characteristics are

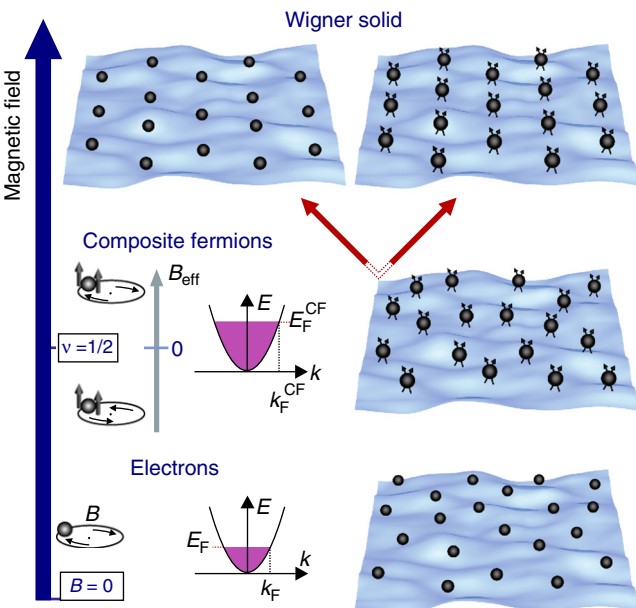

**Fig. 1** Schematic of the phases of a 2DES in a magnetic field: The different phases of a two-dimensional electron system (2DES) in a magnetic field. At zero magnetic field (bottom panel) the electrons are described as a weakly interacting Fermi gas with a well-defined Fermi surface. In the half-filled lowest LL, e.g., at filling factor $\nu = 1/2$, the electrons reduce their mutual interaction by attaching the two magnetic flux quanta, resulting in the emergence of new particles, so-called composite fermions (middle panel)[2,3]. These particles form a Fermi surface at $\nu = 1/2$ and move in an effective field $B_{\text{eff}} = B - B_{\nu=1/2}$ giving rise to magnetoresistance oscillations known as the fractional quantum Hall effect (Fig. 2). At even lower filling factors, a Wigner solid, a crystalline phase of electrons arranged by the repulsive Coulomb force and another manifestation of many-body correlations, becomes the ground state, which can be formed either by bare electrons (top left) or composite fermions (top right)

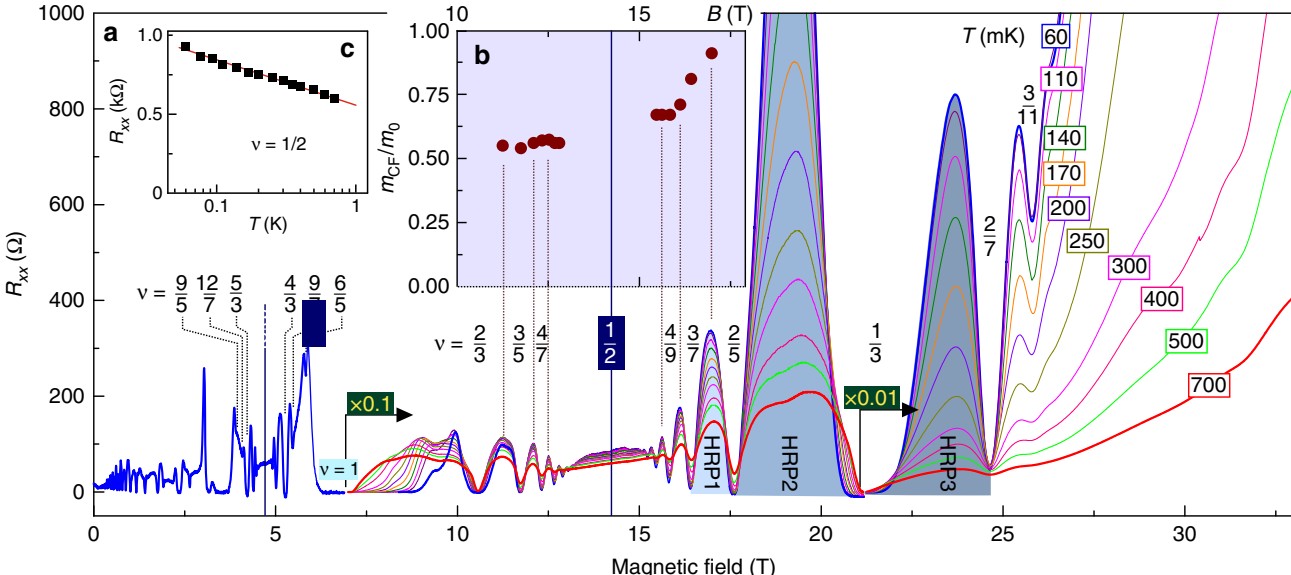

**Fig. 2** Temperature-dependent magnetotransport up to 33 T. **a** $R_{xx}$ at $T = 60$ mK (blue trace). The other colors show $R_{xx}(B)$ in the fractional quantum Hall regime for higher temperatures. The high resistivity phases HRP1, HRP2, and HRP3 are indicated by the shaded region. **b** Mass of composite fermions extracted from the temperature dependence of the $R_{xx}$ oscillation amplitude. **c** Temperature dependence of $R_{xx}$ at $\nu = 1/2$. The decreasing resistance with increasing temperatures indicates a residual interaction between composite fermions

associated with the depinning of the Wigner solid from the disorder, when a certain threshold force is exceeded, and its subsequent sliding along the disorder landscape (Supplementary Fig. 1b). Thus, the trace of a Wigner solid formation appears already in HRP1 between $\nu = 3/7$ and $\nu = 2/5$, whereas a larger $R_{xx}$ and I–V non-linearity at HRP2 and HRP3 indicate a stronger pronounced Wigner solid character.

A non-linear I–V characteristics can also appear in structures with domain boundaries originating from the sample inhomogeneity. To reduce the effect of sample inhomogeneity the structures were being rotated during the molecular epitaxial growth and we used a piece of structure next to the center of the ZnO substrate to ensure a higher sample homogeneity[30]. Although we cannot exclude the effect of sample inhomogeneity on the sample characteristics, our structure demonstrates magnetotransport of quality comparable with the best GaAs structures. Furthermore, as we will demonstrate later in the paper, the effect of in-plane magnetic field on both the high resistive phases and the transport around $\nu = 1/2$ has a largely intrinsic origin.

HRP1 represents an interesting region. While it shows the features of an emerging Wigner solid, at the same time it can be comprised as a part of the $R_{xx}$ oscillations caused by the CFs' orbital motion in $B_{eff}$, and therefore can also be attributed to the liquid phase. We note that the $R_{xx}$ amplitude at HRP2 shoots up so high that it can hardly be associated with a gradual increase of the Shibnikov-de Haas oscillations amplitude with an increasing magnetic field. The estimate of the CF mass can be valuable to characterize the CF liquid phase and to elucidate the characteristics of region HRP1. Thus, we now analyze the CF mass $m_{CF}$ around $\nu = 1/2$ from the temperature dependence of the $R_{xx}$ oscillation amplitude by using the Lifshitz–Kosevich formalism (Supplementary Note 2). Figure 2b displays $m_{CF}$ around $\nu = 1/2$, which extends the linear dependence of $m_{CF}$ on $B$ (Supplementary Fig. 3) to higher field as $m_{CF}/m_0 = 0.047B/\text{T}$[25]. More noticeable is the increase of $m_{CF}$ over the linear trend when the 2DES approaches the high resistivity phase HRP1.

The increase of the composite fermion mass is the signature of an enhanced electron–electron interaction, effectively strengthened by

the reduction of kinetic energy, which is brought about by the electron localization by the magnetic field and/or disorder. An enhanced mass can also possibly be anticipated when CFs become more inert due to their localization. Both scenarios for mass increase signal the formation of a solid state, which is consistent with observing the traces of the Wigner solid character at HRP1, and the more strongly pronounced Wigner solid-like character at HRP2. Our transport experiment cannot distinguish whether the high resistance phase HRP2 is formed by the electron or composite fermion. Following the recent theory[19] HRP2 should correspond to an electron crystal. Then the transport characteristics described above can be treated as experimental attributes of the intricate phase transition between the composite fermion liquid phase and electron solid. HRP3 can potentially be attributed to a two-flux composite fermion Wigner crystal, as it forms in the LL filling factors attributed to a four-flux composite fermion liquid.

**Experiment in tilted magnetic field.** The transport properties discussed above change dramatically when the sample is rotated in the magnetic field, that is, when an additional field component is applied parallel to the 2DES. Since the electron spin susceptibility for this structure is about 2, the opposite spin orientation branch of the lowest LL lies energetically high and is not populated[31]. Thus the spin effects are not anticipated to play a role for the discussion below. Figure 3a depicts $R_{xx}$ traces at several sample orientations $\theta$ obtained at base temperature and shows the asymmetrical impact of the in-plane field on the transport for $\nu < 1/2$ and $\nu > 1/2$ ($\theta$ is the tilt angle between the normal of the 2DES plane and the magnetic field direction). Firstly, one notices that $R_{xx}$ of HRP1, HRP2, and HRP3 increases gradually with an increasing $\theta$, while $R_{xx}$ minima at $\nu = 3/7$, 2/5, and 1/3 do not change significantly. Thus the insulating phase becomes more pronounced by applying an in-plane field. The temperature dependence of $R_{xx}$ maxima of the high resistivity phases is depicted for three representative $\theta$s in Fig. 3b. Furthermore, $R_{xx}$ around $\nu = 1/2$ gains a background, which becomes larger with the increasing $\theta$. Since magnetotransport experiments in GaAs demonstrate the extension of the tail of the insulating phase into

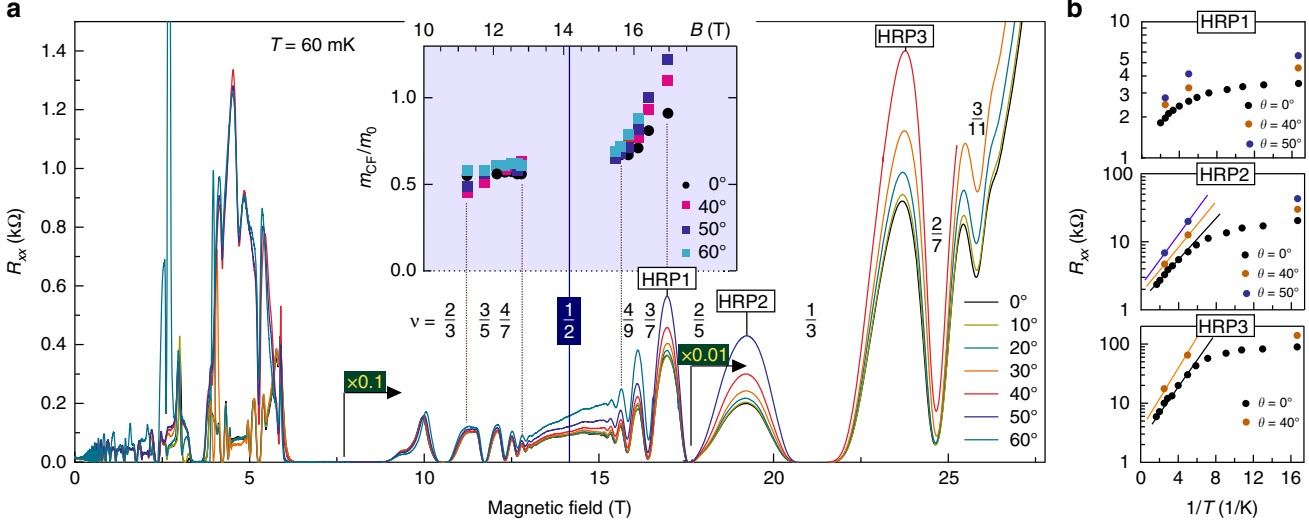

**Fig. 3** Magnetotransport in tilted magnetic fields. **a** The resistance of high resistivity phases HRP1, HRP2, and HRP3 increases with the application of in-plane magnetic field. The insulating phase shifts towards higher filling factors with increasing in-plane field, as can be seen from the growing background around $\nu = 1/2$. Inset: mass of composite fermions evaluated from temperature dependence of $R_{xx}$ oscillation amplitude. **b** Temperature dependence of high resistivity phases HRP1, HRP2, and HRP3 at several tilt angles

the $\nu = 1/2$ region with an increasing $\theta$[32,33], we may also suppose that the background forming around $\nu = 1/2$ has the same origin and is associated with the development of the insulating phase growing towards $\nu = 1/2$ and above.

## Discussion

A strong in-plane field couples to the orbital motion and enhances the inter-subband scattering rate[34]. This will result in an additional LL broadening and should be particularly effective in our ZnO heterostructure, since the subband separation in the confinement potential is of the order of 5 meV and the electron cyclotron energy reaches 3.8 meV at 10 T[35]. The scenario of scattering rate enhancement at high $\theta$ is not consistent with our experimental data. Indeed, for all tilt angles $\theta$ the $R_{xx}$ oscillations are not damped but rather persist on top of an increasing background. Both the field where $R_{xx}$ oscillations onset and the number of $R_{xx}$ oscillations do not change with an increasing in-plane field strength, whereas an anticipated broader LL ought to smear out the oscillations and to increase the field where $R_{xx}$ oscillations set on.

Furthermore, we analyze the temperature dependence of the $R_{xx}$ oscillation amplitude and estimate $m_{CF}$ around $\nu = 1/2$ for several $\theta$s (Supplementary Note 2). The inset of Fig. 3 depicts the result of this analysis. For $\nu > 1/2$, $m_{CF}$ does not show any noticeable change, but it shows a pronounced field and tilt angle dependence for $\nu < 1/2$, that is, for a given perpendicular magnetic field $m_{CF}$ is heavier at a larger tilt angle. The quantum scattering time traced as a fitting parameter in the estimation of $m_{CF}$ does not show a tilt angle dependence around $\nu = 1/2$ (Supplementary Fig. 5). This strengthens our concept of negligible change of inter-subband scattering rate due to the application of in-plane magnetic field. The estimation of scattering time becomes more uncertain at approaching the HRP1 phase. This points to an inadequate description of high resistivity phases with Lifschitz–Kosevitch approximation, possibly supporting our point that the HRP1 phase has a mixed character of state of matter. Therefore the experimental finding implies that the change in magnetotransport caused by the in-plane field application has an intrinsic origin.

The application of an in-plane field can significantly disturb the electron/CF Fermi contour making it elliptic shaped[36–40]. The effective particle mass is now given by $m* = \sqrt{m_t \times m_l}$, where $m_t$ and $m_l$ are the masses along two prime axises of the ellipse. Such anisotropy might be responsible for the CF mass enhancement beyond the mass estimated at zero tilt angle. An enhanced mass can also signal that the electron–electron interaction becomes stronger, which brings the system closer to the condition to form a crystalline phase.

In order to further address the in-plane field induced stabilization of the Wigner solid-like phase we now analyze how much the in-plane field squeezes the electron wavefunction width, as it effectively enhances the Coulomb interaction and can affect the transport properties[41–43]. In zero in-plane field the wavefunction width of the heterostructure is about 10 nm wide[31,35]. At $\theta = 50°$ it is squeezed down to 2.6 nm at $B_\perp = 12$ T, representing the region $\nu > 1/2$, and down to 2.2 nm at $B_\perp = 17$ T, representing the region $\nu < 1/2$ (Supplementary Note 3). Since the wavefunction width reduces significantly with in-plane field on both sides of $\nu = 1/2$ compared with zero in-plane field, the Coulomb interaction should also be commensurately enhanced around $\nu = 1/2$. Nonetheless, $m_{CF}$ defined by the interaction effects remains almost unchanged for $\nu > 1/2$ and no large effect of in-plane field on transport characteristics is seen in this region. Consequently, the increase of $m_{CF}$ for $\nu < 1/2$ is not mainly caused by the reduced wavefunction width. However, a smaller width of the wavefunction favors the solid phase over the liquid phase[19]. This can explain an increased resistivity of HRP1, HRP2, and HRP3 phases and the gradual shift of insulating phase towards higher filling factors. In this scenario, the $m_{CF}$ increase in $\nu < 1/2$ region is consistent with our observation that $m_{CF}$ is larger the closer the electron system approaches the high resistivity phase, which may be interpreted as an effective localization of the CF.

Finally, we discuss the filling factor $\nu = 1/2$, which is originally thought to have a Fermi surface of CF (Fig. 1). Figure 4 depicts the temperature dependence of $R_{xx}$ at several $\theta$'s. At $\theta = 0$, considering the presence of only CF Fermi surface, the logarithmic temperature dependence of $R_{xx}$ points toward a residual CF interaction[2,25,44]. At larger $\theta$ the slope of logarithmic $R_{xx}$ temperature dependence increases, pointing to an increased CF residual interaction. On the other hand, our experiment demonstrates the shift of the insulating phase towards higher filling factors with an increasing $\theta$. Consequently, the state

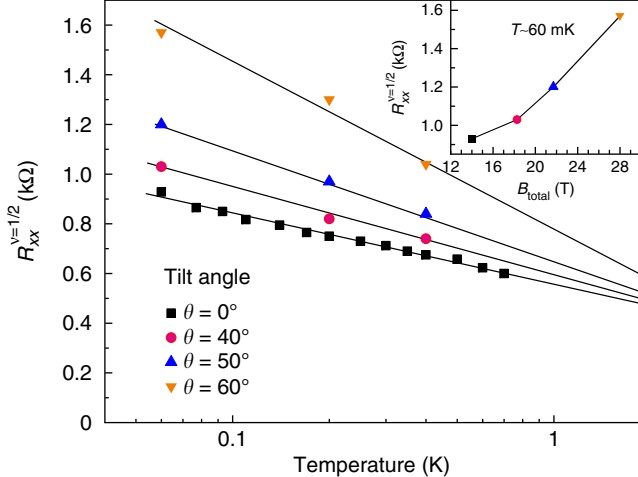

**Fig. 4** $R_{xx}$ temperature dependence at $\nu = 1/2$. The resistance decreases with increasing temperature indicating a residual interaction between the composite fermions at zero tilt angle. The slope becomes more pronounced at higher tilt angles, i.e., a stronger in-plane field, and points towards a more robust Wigner solid phase

at $\nu = 1/2$ is formed by both a CF Fermi liquid with a Fermi surface and tail of the insulating phase forming in the course of liquid-solid phase transition. Then the $R_{xx}$ temperature dependence at each $\theta$ in Fig. 4 reflects not only the CF residual interaction but also the melting of the intermediate liquid-solid phase growing towards $\nu = 1/2$. Giving a small $R_{xx}$ increase between $\theta = 0$ and $\theta = 40$ the intermediate phase is absent or its contribution is small at $\nu = 1/2$ for $\theta = 0$. Our consideration raises questions about the stability of a CF Fermi surface exposed to a strong in-plane magnetic field, whereas the 2DES characteristics in a tilted magnetic field can likely be the attributes of the precursor for the new insulating state in a strong in-plane magnetic field proposed by Piot et al.[33].

Our experimental data show that the electron system enters a correlation regime, which reflects the character of both solid and liquid phases for $\nu < 1/2$ and can likely be the features of a nontrivial phase transition in the electron system occupying the lowest LL. Our experimental results are interpreted within the CF approach, which has recently attracted renewed attention from theory predicting that the CF can be Dirac particles[45–48]. This also introduces an exciting perspective for ZnO studies. Our experimental results display the composite fermion paradigm in a system distinct from conventional semiconductor systems and address the question on how the charge carrier system translates between liquid and solid phases.

## Methods
**Sample**. The sample under study is a MgZnO/ZnO heterostructure (a sequence of about 500-nm thick epitaxial ZnO film (buffer layer) and 250-nm thick MgZnO (capping layer) grown on ZnO substrate) with a charge carrier density $n = 1.7 \times 10^{11}$ cm$^{-2}$ and a mobility $\mu = 600{,}000$ cm$^2$V$^{-1}$s$^{-1}$ at the base temperature of our dilution refrigerator $T = 60$ mK. The LL mixing for this charger carrier density is $\kappa = 4.2$ at $\nu = 1/2$.

**Magnetotransport in tilted magnetic field**. The sample is mounted on a rotating stage allowing in-situ sample rotation in the magnetic field. The tilt angle $\theta$ is determined from the shift of $R_{xx}$ resistance minima of the well-known fractional quantum Hall states.

## Data availability
The data that support the findings of this study are available from the corresponding author upon request.

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

## Acknowledgements

We acknowledge the support of the HFML-RU/FOM member of the European Magnetic Field Laboratory (EMFL). This work was partly supported by JST CREST No. JPMJCR16F1. We would like to thank M. Kawamura, A. S. Mishchenko, M. Ueda, K. von Klitzing, N. Nagaosa, and J. K. Jain for fruitful discussions and comments.

## Author contributions

D.M. and M.K. initiated the project. M.K. supervised the project. D.M. conceived and designed the experiment. J.F. and Y.K. fabricated the samples. D.M., A.M., J.B., and U.Z. performed the high-field magnetotransport experiment and analyzed the data. D.M. and U.Z. wrote the paper with input from all co-authors.

## Additional information

**Competing interests:** The authors declare no competing interests.

