## [Peer Review File · Nature Communications]

Editorial Note: This manuscript has been previously reviewed at another journal that is not operating a transparent peer review scheme. This document only contains reviewer comments and rebuttal letters for versions considered at Nature Communications. Mentions of prior referee reports have been redacted.

REVIEWERS' COMMENTS:

Reviewer #1 (Remarks to the Author):

I have thoroughly read the authors' response to all referees, as well as their resubmitted manuscript. I would like to stress again the most important facts (in my view) regarding this work:

i) absolutely state-of-the-art transport data taken in a new 2DEG system. By new here, I mean not GaAs, nor graphene. I am familiar with both latter systems, albeit more the ultra-high mobility GaAs designed structures, and I shall say that the data presented in this work is truly impressive. Perhaps I am mistaken, but the "wavefunction" engineering that has taken years for the community in GaAs to fully grasp, here is in this work -and I weight in my words- a true gift from exploratory material sciences. I commend all authors for pursuing that route.

ii) A possible transition from CFs physics to localization that in my knowledge has never truly been observed before. As in all papers invoking word such as "anyons", "Composite Fermions", there are some potential criticism simply because those are not nailed down problems. Here, what is new, is that the 2DEG located at the interface has given risen to something truly new. I am not fully sure why, but here it has. That the system underwent localization, I accept it. Which kind? This is s hard question, but this is also *true* for so many other words. Upon re-reading the new draft, I came to the conclusion that it is not a mere remake of the what we've done in GaAs, but rather that it is a serious leap forward coming from a different, perhaps unexpected angle.

My understanding of Nature comm 's mission is to publish work that proposes "paradigm shif" of broad interest (if I understand it well). While this work will not replace immediately GaAs or Graphene, it is certainly a serious potential game changer that had not been trivially spotted.

I therefore wholeheartedly recommend publication based on the arguments that are mentioned above.

Reviewer #2 (Remarks to the Author):

I think the authors have given good answers to my points, and to those of other reviewers.

I am grateful for the authors' point concerning Ref 22, as explained in their reply, that high- LL-mix solid can be favored by more wave function compression. But it is not clear clear to me if that effect is alone shifting solids to higher ν , or if a 3d Piot type thing is going on, or both in combination.

I regard this as a minor point, the main one is the observation of the many insulators, which are well-interpreted by the authors in light of current state of theory, and I favor publication.

minor: Use of the word "intrinsic" on p 9, beginning. Does this mean not having to do with scattering or disorder here?

trivial: "Owing to a simple band structure " not "owned to".

Reviewer #3 (Remarks to the Author):

Second report on the manuscript NPHYS-2017-07-02214 "Phase transition from a composite fermion liquid to a Wigner solid in the lowest Landau level of ZnO" by D. Maryenko et al.

The authors have considerably improved their interesting manuscript in response to the first round of referee comments. In particular, the experimental results are now compared to the new Ref. [22], a recent theoretical study of the competition between composite fermion liquids and Wigner solids. All my questions and remarks have been answered, and where appropriate the authors modified their manuscript. As a result, the authors' conclusions are now well supported by the careful analysis of beautiful experimental data, and I warmly recommend the manuscript for publication.

I have a suggestion for improving the accessibility of the presentation: in Figs. 2 and 3, it is easy to overlook the inserts for $\times 0.1$ and $\times 0.01$ reduction of resistance. It might be helpful to make them more visible, and/or explicitly mention them in the respective captions.

Reviewer #1 (Remarks to the Author):

I have thoroughly read the authors' response to all referees, as well as their resubmitted manuscript. I would like to stress again the most important facts (in my view) regarding this work:

i) absolutely state-of-the-art transport data taken in a new 2DEG system. By new here, I mean not GaAs, nor graphene. I am familiar with both latter systems, albeit more the ultra-high mobility GaAs designed structures, and I shall say that the data presented in this work is truly impressive. Perhaps I am mistaken, but the "wavefunction" engineering that has taken years for the community in GaAs to fully grasp, here is in this work -and I weight in my words- a true gift from exploratory material sciences. I commend all authors for pursuing that route.

ii) A possible transition from CFs physics to localization that in my knowledge has never truly been observed before. As in all papers invoking word such as "anyons", "Composite Fermions", there are some potential criticism simply because those are not nailed down problems. Here, what is new, is that the 2DEG located at the interface has given risen to something truly new. I am not fully sure why, but here it has. That the system underwent localization, I accept it. Which kind? This is a hard question, but this is also *true* for so many other words. Upon re-reading the new draft, I came to the conclusion that it is not a mere remake of the what we've done in GaAs, but rather that it is a serious leap forward coming from a different, perhaps unexpected angle.

My understanding of Nature Comm 's mission is to publish work that proposes "paradigm shift" of broad interest (if I understand it well). While this work will not replace immediately GaAs or Graphene, it is certainly a serious potential game changer that had not been trivially spotted.

I therefore wholeheartedly recommend publication based on the arguments that are mentioned above.

We would like to thank the Reviewer for the detailed and critical examination of our manuscript. We are particularly grateful to the Reviewer for identifying that our work may offer a new perspective on studying the correlation phases in two-dimensional system and for recommending our work for publication in Nature Communication.

Reviewer #2 (Remarks to the Author):

I think the authors have given good answers to my points, and to those of other reviewers.

I am grateful for the authors' point concerning Ref 22, as explained in their reply, that high- LL- mix solid can be favored by more wave function compression. But it is not clear to me if that effect is alone shifting solids to higher ν , or if a 3d Pirot type thing is going on, or both in combination.

I regard this as a minor point, the main one is the observation of the many insulators, which are well-interpreted by the authors in light of current state of theory, and I favor publication.

We are grateful to the Reviewer for a careful examination of our manuscript and for

recommending it for publication. We agree with the Reviewer that there are still open questions about the shift of insulating phase to higher filling factors. We believe this can stimulate further theoretical and experimental works, that will use alternative measurement techniques and will not be limited only to ZnO.

minor: Use of the word “intrinsic” on p 9, beginning. Does this mean not having to do with scattering or disorder here? trivial: “Owing to a simple band structure “ not “owned to”.

Our experimental results and their analysis suggests that the effect of the in-plane magnetic field on the magnetotransport is not the result of disorder change. We therefore identify that the change in the magnetotransport caused by the in-plane field has intrinsic origin.

We would like to thank the Reviewer for pointing out the misspelling. We corrected “Owing to...” to “Owned to.... ”.

Reviewer #3 (Remarks to the Author):

Second report on the manuscript [Redacted] "Phase transition from a composite fermion liquid to a Wigner solid in the lowest Landau level of ZnO" by D. Maryenko et al.

The authors have considerably improved their interesting manuscript in response to the first round of referee comments. In particular, the experimental results are now compared to the new Ref. [22], a recent theoretical study of the competition between composite fermion liquids and Wigner solids. All my questions and remarks have been answered, and where appropriate the authors modified their manuscript. As a result, the authors' conclusions are now well supported by the careful analysis of beautiful experimental data, and I warmly recommend the manuscript for publication.

I have a suggestion for improving the accessibility of the presentation: in Figs. 2 and 3, it is easy to overlook the inserts for x0.1 and x0.01 reduction of resistance. It might be helpful to make them more visible, and/or explicitly mention them in the respective captions.

We would like to thank the Reviewer for careful and positive examination of our work recommending it for publication.

To improve the accessibility of Figures 2 and 3 we highlighted the change of scaling factor x0.1 and x0.01.